# The Microbiota Profile in Inflamed and Non-Inflamed Ileal Pouch–Anal Anastomosis

**DOI:** 10.3390/microorganisms8101611

**Published:** 2020-10-20

**Authors:** Sabrina Just Kousgaard, Thomas Yssing Michaelsen, Hans Linde Nielsen, Karina Frahm Kirk, Mads Albertsen, Ole Thorlacius-Ussing

**Affiliations:** 1Department of Gastrointestinal Surgery, Aalborg University Hospital, Hobrovej 18-22, 9000 Aalborg, Denmark; otu@rn.dk; 2Department of Clinical Medicine, Aalborg University, Søndre Skovvej 15, 9000 Aalborg, Denmark; halin@rn.dk; 3Center for Microbial Communities, Aalborg University, Frederik Bajers Vej 7H, 9220 Aalborg, Denmark; tym@bio.aau.dk (T.Y.M.); ma@bio.aau.dk (M.A.); 4Department of Clinical Microbiology, Aalborg University Hospital, Mølleparkvej 10, 9000 Aalborg, Denmark; 5Department of Infectious Disease, Aalborg University Hospital, Mølleparkvej 4, 9000 Aalborg, Denmark; kfk@rn.dk

**Keywords:** *Bacteroides*, dysbiosis, *Escherichia*, *Enterobacteriaceae*, FAP, IPAA, microbiota, pouchitis

## Abstract

The objective was to determine the bacterial composition in inflamed and non-inflamed pouches for comparison to the microbiota of healthy individuals. Pouch patients and healthy individuals were included between November 2017 and June 2019 at the Department of Gastrointestinal Surgery, Aalborg University Hospital, Denmark. A faecal sample was collected from all participants for microbiota analysis using 16S rRNA amplicon sequencing. Overall, 38 participants were included in the study. Eleven patients with a normally functioning pouch, 9 patients with chronic pouchitis, 6 patients with familial adenomatous polyposis, and 12 healthy individuals. Patients with chronic pouchitis had overall lower microbial diversity and richness compared to patients with a normal pouch function (*p* < 0.001 and *p* = 0.009) and healthy individuals (*p* < 0.001 and *p* < 0.001). No significant difference was found between patients with familial adenomatous polyposis and chronic pouchitis (microbial diversity *p* = 0.39 and richness *p* = 0.78). Several taxa from the family *Enterobacteriaceae*, especially genus *Escherichia*, were associated primarily with patients with chronic pouchitis, while taxa from the genus *Bacteroides* primarily were associated with healthy individuals and patients with a normally functioning pouch. Finally, a microbial composition gradient could be established from healthy individuals through patients with normal pouch function and familial adenomatous polyposis to patients with chronic pouchitis.

## 1. Introduction

Chronic pouchitis is the primary reason for long-term functional disturbance for patients with an ileal pouch–anal anastomosis (IPAA) [1,2]. Pouchitis is unusual in familial adenomatous polyposis (FAP) patients with pouches, but occurs in up to 60% of patients with a pouch created in the surgical treatment of ulcerative colitis (UC) [3,4].

The pathogenesis of pouchitis remains uncertain, although many theories have been suggested. Current evidence suggests that the gut microbiota is a major factor in the aetiology of pouchitis [5]. This hypothesis is supported by clinical observations of the symptomatic effect of antibiotic treatment. However, after initial response to antibiotics, more than half of the patients have recurring episodes of pouchitis and about 5% of the patients develop chronic pouchitis [4,6]. Simultaneously, it is likely that there is an abnormal activation of the immune system in pouchitis, caused by yet unidentified factors. The initial success of antibiotics in the treatment of pouchitis also indicates that gut bacteria is a likely trigger, activating the mucosal immune system [7].

Probiotics and faecal microbiota transplantation have been found to decrease the rate of pouchitis, indicating that dysbiosis of the microbiota is an imperative factor [8,9,10]. However, direct evidence of the role of a dysbiotic microbiota in the pathogenesis of pouchitis is missing [11]. Studies investigating the composition of the bacteria in the gut are often based on traditional culture techniques, which will detect less than 50% of bacteria in the lumen depending on the culture medium and cultivation method [12]. Use of molecular techniques for DNA-based identification of the composition of the microbiota has provided researchers with the opportunity to evaluate the microbiota profile in the pouch more effectively [13].

The aim of our study was to determine the bacterial composition of inflamed and non-inflamed pouch microbiota and compare it to the microbiota of healthy individuals, using bacterial microbiota profiling.

## 2. Materials and Methods

### 2.1. Trial Design

Pouch patients and healthy individuals were included in a single-centre, cross-sectional study. A faecal sample was collected from all patients and healthy controls (HCs).

The primary endpoint was to determine the bacterial composition of pouch microbiota in inflamed and non-inflamed pouches by bacterial microbiota profiling. The secondary endpoint was to compare pouch microbiota to the microbiota of HCs.

### 2.2. Participants

Patients and HCs were recruited between November 2017 and June 2019 at the Department of Gastrointestinal Surgery, Aalborg University Hospital, Aalborg, Denmark. The participants were contacted by telephone for recruitment, or during outpatient visits.

Asymptomatic patients with a normally functioning pouch (no symptoms of pouchitis) after IPAA surgery for UC were identified using patient records at the hospital. Inclusion criteria for patients with a normal pouch function were the absence of documented episodes of pouchitis, absence of symptoms of pouch dysfunction, and no use of antibiotics for pouchitis within the last years prior to inclusion.

Patients with chronic pouchitis and patients with FAP were identified using patient records at the hospital. Chronic pouchitis was defined as ≥3 episodes of pouchitis diagnosed according to clinical symptoms, endoscopic signs of inflammation and histologic inflammation of pouch biopsies with a Pouchitis Disease Activity Index (PDAI) score ≥7 within the last year [14]. Patients with chronic pouchitis were also included in a pilot study using faecal microbiota transplantation in the treatment of pouchitis, as described by Kousgaard et al. [15]. Faecal samples were collected before study intervention in the pilot study. Patients with FAP had previously undergone surgery with removal of the large intestine after diagnosis of FAP. The FAP patients had no history of pouchitis and no use of antibiotics for pouchitis within the last years prior to inclusion.

Overall, patients were grouped according to inflammation (patients with chronic pouchitis) or no inflammation (patients with a normal pouch function and FAP) of the pouch.

HCs without disease in the colon or rectum were recruited from the Blood Bank at Aalborg University Hospital, Aalborg, Denmark.

Each participant was asked to deliver a faecal sample when included in the study. The participants also completed a questionnaire to obtain information about daily bowel movements, use of any type of antibiotics within the last 6 months (including specific use of antibiotics for pouchitis (ciprofloxacin and/or metronidazole) and continuous use of antibiotics) and information about current pouch function for the pouch patients (using the clinical PDAI score, cPDAI).

### 2.3. Sample Preparation

A faecal sample was collected from all patients and HCs for microbiota analysis.

All samples were stored in a biobank at minus 80 degrees before further tests. DNA was extracted from the faecal samples using QIAamp PowerFecal DNA Kit (QIAGEN, Copenhagen, Denmark) according to the manufacturer’s instructions. Bacterial microbiota profiling (the hypervariable V4-region of the 16S rRNA gene) was used to analyse microbiota in the faecal samples.

DNA was prepared for sequencing by a two-step PCR amplification. The first PCR amplification was prepared as 25 µL reactions using PCRBIO Ultra Mix (PCR Biosystems, localities) with 10 ng of isolated DNA as template and 400 nM primer mix (515F: GTGYCAGCMGCCGCGGTAA [16] and 806R: GGACTACNVGGGTWTCTAAT [17]). Thermocycler settings for the first PCR: Initial denaturation at 95 °C for 2 min, 30 cycles of 95 °C for 15 s, 55 °C for 15 s, 72 °C for 50 s, and a final elongation for 5 min. Both a negative and a positive control were included in the PCR setup. The negative control consisted of nuclease-free water. The positive control contained template DNA from an anaerobic digester sample known to amplify PCR product with the selected primer set. The first PCR amplification was performed with duplicate reactions for each sample and pooled after amplification. Incorporation of barcodes was carried out in a second PCR amplification. The reactions (25 µL) were performed with 2 µL cleaned amplicon PCR product (diluted to 5 ng/µL) as template, as well as X5 PCRBIO reaction buffer (×1), PCRBIO Hifi polymerase (1U) and 1 µM Illumina adaptor mix. Thermocycler settings for the second PCR: Initial denaturation at 95 °C for 2 min, 8 cycles of 95 °C for 20 s, 55 °C for 30 s, 72 °C for 60 s and a final elongation at 72 °C for 5 min. The second PCR was performed in single reactions. PCR product from both PCR runs was purified using 0.8× CleanNGS beads (CleanNA) and eluted in nuclease-free water. DNA concentrations were measured with Quant-iT HS DNA Assay (Thermo Fisher Scientific) and the purified PCR amplicon products were visualised on D1K ScreenTapes using a TapeStation 2200 Analyzer (Agilent). Prior to sequencing, all samples were pooled into one tube in equimolar concentrations and barcoded with the Nextera indexing kit. Sequencing of the library pools was performed on the Illumina MiSeq platform with v3 chemistry and 2 × 301 bp paired-end setting.

### 2.4. Data Analysis

The raw sequencing data were summarised into amplicon sequencing variants (ASVs) using an in-house pipeline AmpProc v5.1 (http://www.github.com/eyashiro/AmpProc/), primarily using the USEARCH v10.0.240 workflow [18]. The ASVs were assigned taxonomy using SILVA LTP vers. 132 as reference database (https://www.arb-silva.de/) [19]. The raw sequencing data were demultiplexed with bcl2fastq v2.17.1.14 then processed with AmpProc v5.1beta1.0 (http://www.github.com/eyashiro/AmpProc/), which is primarily based on the USEARCH v10.0.240 workflow [18]. Data were summarised into ASVs to avoid clustering by nucleotide identity to maximise taxonomic resolution and reduce clustering biases [20]. After filtering out phix contamination, the paired-end reads were merged using –fastq_mergepairs with settings “-fastq_maxdiffs 15” [21]. Merged reads were filtered to determine quality using –fastq_filter with settings “-fastq_maxee 1 –fastq_minlen 200”, dereplicated using –fastx_uniques and clustered into ASVs by –unoise3. Clustered reads were further curated by filtering out reads more distantly related to known sequences than 60% identity using –userarch_global with settings “-db gg_13_8_otus97/97_otus-fasta –id 0.6 –maxaccepts 1 –maxrejects 8” [22]. The ASV table was generated using the function –otutab with settings “–zotus and –id 0.97” [23], and taxonomy was assigned to each ASV using –sintax with setting “-sintax_cutoff 0.8” along with the SILVA LTP vers. 132 as the reference database (https://www.arb-silva.de/) [19].

Data analysis was performed in R v. 3.6.0 through Rstudio v. 1.1.383 (http://www.rstudio.com) primarily using the packages ampvis2, tidyverse, and vegan [24,25,26]. Community richness was calculated using an observed number of ASVs and diversity was calculated using the Shannon index. Samples were rarefied to the lowest observed sequencing depth (16,245 reads) for richness and diversity estimates. Beta diversity was examined using principal component analysis (PCA) on Hellinger transformed ASV abundances. Filtering of ASVs with low variance, defined as >50% of samples associated with one value, was performed prior to PCA. To assess the statistical significance of the groupings in PCA, permutation tests of pairwise linear regression were performed using the pairwise.factorfit function from the RVAideMemoire package [27]. For statistical comparison, the Wilcoxon rank sum test was used, and the Holm *p*-value correction was used to address multiple testing [28]. An adjusted *p*-value < 0.05 was considered statistically significant.

### 2.5. Ethics

The study was performed with the requirements of Good Clinical Practice and the Revised Declaration of Helsinki. All participants provided signed written informed consent to participate in the study. The Regional Research Ethics Committee of Northern Jutland, Denmark approved the study (N-20180013), 3 April 2018. Five of the HCs were included in connection with another study (N-20150021, approved 1 June 2015).

## 3. Results

### 3.1. Patient Population

Overall, 38 participants were included in the study, 11 patients with normal pouch function, 9 patients with chronic pouchitis, 6 patients with FAP but without registered episodes of pouchitis, and 12 HCs. Participant characteristics are summarised in Table 1.

### 3.2. Analysis of Gut Microbiota

Sequencing was successful for all samples and produced satisfactory reads (16,245 to 28,512 reads) to cover the community for all patients and HCs as determined by rarefaction analysis. Sequencing data are deposited in the Sequencing Read Archive https://www.ncbi.nlm.nih.gov/sra/ (accession number: PRJNA646261).

Faecal samples collected from patients with chronic pouchitis had lower microbial diversity and richness compared to patients with normal pouch function (*p* < 0.001 and *p* = 0.009) and HCs (*p* < 0.001 and *p* < 0.001) (Figure 1). Faecal samples from patients with FAP did not significantly differ in microbial diversity or richness compared to patients with chronic pouchitis (*p* = 0.39 and *p* = 0.78) (Figure 1). In general, microbial diversity and richness were also lower in patients with normal pouch function compared to HCs without disease in the colon or rectum (*p* = 0.005 and *p* = 0.003) (Figure 1).

Relative abundance of bacterial species between the three groups of patients and HCs showed that patients with normal pouch function and HCs overall had a similar composition of their microbiota (Figure 2A). For most of the patients with a normally functioning pouch and HCs, the most abundant genera were the genus *Bacteroides*, or the genus *Prevotella* for the remaining few (Figure 2A). Subsequently, the genus *Faecalibacterium* or *Dialister* from the phylum *Firmicutes* was the most abundant genera in patients with normal pouch function and HCs (Figure 2A). Moreover, the composition of the microbiota in patients with normal pouch function and HCs was more homogenic compared to patients with chronic pouchitis and FAP. For patients with chronic pouchitis or FAP, the genus *Bacteroides* was most prevalent in some patients, where for others the genus was not detected as one of the 20 most abundant genera (Figure 2A). The two chronic pouchitis patients (patient ID: Pouchitis_001 and Pouchitis_005) with no detection of the genus *Bacteroides* had both received continues long-term treatment with ciprofloxacin and metronidazole (Figure 2A). The remaining chronic pouchitis patients (patient ID: Pouchitis_007) receiving continuous antibiotics had *Bacteroides* as the most prevalent genera. All the included patients with chronic pouchitis had received one or several antibiotic treatments for pouchitis, within the year up to study inclusion.

Throughout the principal component analysis (PCA) plot, it was possible to separate HCs from all other groups (*p* < 0.05, Figure 2B). Furthermore, patients with normal pouch function and FAP scattered as intermediate between HCs and patients with chronic pouchitis along PC1, which explained a considerable portion of the variance (38.7%; Figure 2B). Albeit pairwise comparisons of patients with a normal pouch function, FAP and chronic pouchitis were not statistically significant (*p*-value between 0.14 and 0.26), this suggests a gradient in the microbial composition going from HCs through normal pouch function, and FAP to chronic pouchitis. Inspecting the ASVs with heights absolute weights on the PC1 axis (Figure 2C) several ASVs from genus *Bacteroides* were associated primarily with HCs, while ASVs from the family *Enterobacteriaceae*, especially genus *Escherichia* primarily were associated to patients with chronic pouchitis.

## 4. Discussion

This study used bacterial microbial profiling with 16S rRNA amplicon sequencing to determine the bacterial composition of pouch microbiota. In particular, the aim was to elucidate a distinct microbial profile for patients with an inflamed pouch, compared to a pouch without confirmed inflammation.

Overall, we found that patients with an inflamed pouch had lower microbial diversity and richness compared to healthy individuals and patients with a pouch without inflammation. This was previously also described by Landy et al., where pouchitis patients’ stool was characterised by low bacterial richness and diversity, compared with stool samples from healthy individuals [29]. This lower diversity and richness in stool samples from patients with an inflamed pouch compared to non-inflamed pouches could indicate a link between dysbiosis of the microbiota and pouch inflammation. However, it is important to take into consideration that patients with a normal functioning pouch still have lower microbial diversity and richness, compared to healthy individuals with a colon. Furthermore, we found no difference in microbial diversity and richness between patients with FAP and chronic pouchitis patients. This could suggest that factors other than changes in the microbial composition can influence inflammation of the pouch. Finally, frequent bowel movements may impact the microbial profile. A study by Kwon et al. [30] found that microbial richness tended to decrease with increasing number of defecations in a group of healthy individuals.

A recent study by Petersen et al. [31] found that ileo-anal pouch anastomosis patients had an increased abundance of the phylum *Proteobacteria* compared to patients with UC or Crohn’s disease and healthy individuals. Furthermore, a higher abundance of the phylum *Fusobacteria* was found in pouch patients with a faecal calprotectin level above 500 µg/mg as a measure for an inflamed pouch [31]. In our study, the difference of the microbiota of chronic pouchitis patients compared to healthy individuals and patients with a non-inflamed pouch could mainly be explained by several taxa from the genus *Bacteroides*, which were most abundant in healthy individuals and patients with a normal pouch function, while taxa from the family *Enterobacteriaceae* including genus *Escherichia* primarily were most abundant in patients with chronic pouchitis. The abundance of the phylum *Proteobacteria* was increased in both patients with a normal functioning pouch and healthy individuals compared to chronic pouchitis patients. Two out of the three chronic pouchitis patients receiving continued long-term treatment with ciprofloxacin and metronidazole had no detection of members of the *Bacteroides* genera among the most abundant genera, which possibly could be explained by the previous use of antibiotics, as also suggested by Petersen et al. [31].

Another study by Pawełka et al. found that chronic pouchitis or patients in need of long-term antibiotic treatment to control symptoms of pouchitis were associated with a significantly higher numbers of *Staphylococcus aureus* in faecal cultures [32]. Our research group found no difference between the present of *S. aureus* in faecal cultures in patients with chronic pouchitis compared to the Danish background population (data not published). Contradictory to the findings of Pawełka et al. [32], Tannock et al. reported a significantly lower number of *Enterococcus spp*., *Faecalibacterium prausnitzii*, *Clostridium spp*., *Ruminococcus spp*., *Eubacterium spp*., *Lachnospiraceae* and *Insertae Sedis XIV* in chronic pouchitis compared to healthy individuals, by use of faecal cultivation [33].

Furthermore, Landy et al. [29] showed that at the phylum level, pouchitis patients’ stool samples were characterised by a higher proportion of *Proteobacteria* compared to healthy individuals’ stool samples. At the family level, pouchitis patients’ stool samples were characterised by a lower proportion of common obligate anaerobic lineages such as *Ruminococcaceae*, *Coriobacteriaceae*, *Porphyromonadaceae* and *Rikenellaceae*, and higher proportional abundances of *Enterobacteriaceae* and *Clostridiaceae* [29], as also demonstrated in our study. When looking at the genus and 97% operational taxonomic unit levels, Landy et al. found that pouchitis patients’ stool samples contained lower proportional abundances of many obligate anaerobes, including *F. prausnitzii*, and higher proportions of *Escherichia/Shigella spp*. and *Ruminococcus gnavus* compared with healthy individuals’ stool samples [29]. In general, we found that the abundance of several bacterial taxa, including taxa from the phylum *Firmicutes* and *Proteobacteria,* were lower in patients with chronic pouchitis, compared to patients with a normal pouch function and healthy individuals.

The strength of this study is that it compares the composition of the pouch microbiota among several groups of patients and healthy individuals, to describe the differences in microbial profiles of inflamed and non-inflamed pouches. All samples were collected and sequenced according to the same protocol, to limit the risk of bias. Furthermore, the PDAI score, instead of faecal calprotectin levels, was used to identify patients with pouchitis, since PDAI is the recognised scoring system for diagnosing pouchitis and assessing severity of illness [34].

Several limitations need to be addressed. In total, a reasonable number of participants were included, but when subdividing the participants into groups, the sample size of each group is low, which makes it difficult to draw definitive conclusions, especially for FAP patients. Faecal samples from patients with UC or Crohn’s diseases were not included in our study, as it was outside the scope of our aim. However, it would have been relevant to include UC patients with an intact colon, to compare inflamed pouch microbiota with inflamed colon microbiota, as described by Petersen et al. [31]. All patients with chronic pouchitis had received one or several antibiotic treatments before inclusion in the study, as antibiotics are the primary treatment of pouchitis. This will inevitably influence the composition of the patients’ gut microbiota and affect the microbial results. Moreover, other types of medications could also influence the gut microbiota and would have been relevant to include in our study. Microbiome data in our study were generated from 16S amplicon sequencing, which is more informative than using faecal cultures [35]. However, metagenomics would have been the best method to generate the total microbiome data. Our data were summarised into ASVs to avoid clustering by nucleotide identity, in order to maximise taxonomic resolution and reduce clustering biases [20]. In general, caution should be taken when linking changes in microbiome composition to specific ASVs, because the capability to correctly assign taxonomy using 16S amplicon data is still uncertain [36,37]. Finally, the retrospective design of the study allows only association and not causation to be inferred from the results.

In conclusion, patients with an inflamed pouch had lower microbial diversity and richness compared to non-inflamed pouches. Divergent microbial profiles were found between inflamed and non-inflamed pouches, with taxa from the family *Enterobacteriaceae*, especially genus *Escherichia* primarily associated to patients with chronic pouchitis, and taxa from genus *Bacteroides* associated with healthy individuals and patients with a normally functioning pouch. However, no significant difference was found between patients with familial adenomatous polyposis and chronic pouchitis. Future studies should use metagenomic sequencing to further investigate the association between pouch inflammation and the microbiota of the pouch.

## Figures and Tables

**Figure 1 microorganisms-08-01611-f001:**
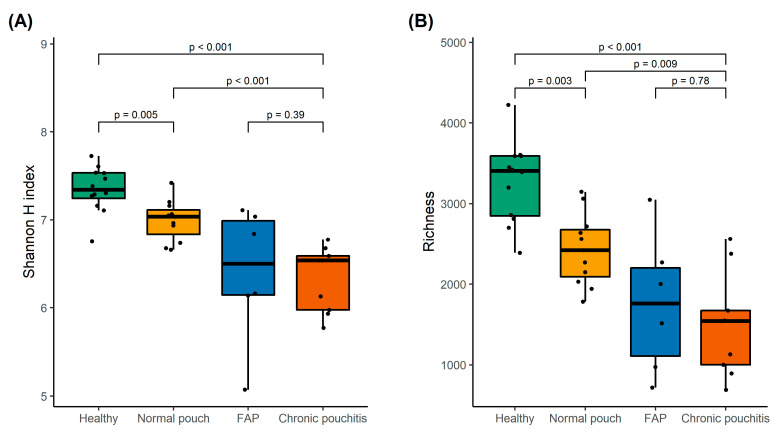
The Shannon diversity index (**A**) and number of amplicon sequencing variants (ASVs) for species richness (**B**) in healthy individuals and patients with normal pouch function, familial adenomatous polyposis (FAP), and chronic pouchitis. *p*-values were calculated using the Wilcoxon rank-sum test and adjusted for multiple comparisons using the Holm’s method.

**Figure 2 microorganisms-08-01611-f002:**
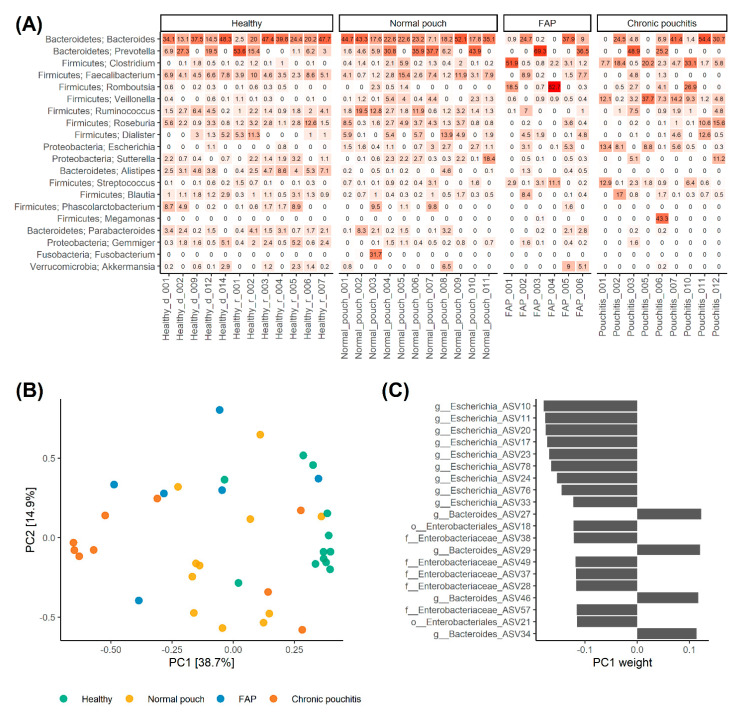
Microbial composition in healthy individuals and patients with normal pouch function, familial adenomatous polyposis (FAP), and chronic pouchitis. The top 20 most abundant genera with phylum names, ordered from top to bottom by mean abundance, are shown for all patients and healthy individuals in (**A**). A principal component analysis (PCA) plot of the first two components for all samples from the patients and healthy individuals is shown in (**B**) and coloured accordingly. In (**C**) are the top 20 most influential amplicon sequencing variants (ASVs) on the first principal component (PC1) ordered from top to bottom by absolute value. Names are shown on the *y*-axis, with corresponding weights on the *x*-axis.

**Table 1 microorganisms-08-01611-t001:** Summary of participant characteristics (*n* = 38).

Group	Normal Pouch (*n* = 11)	Chronic Pouchitis (*n* = 9)	FAP (*n* = 6)	HCs (*n* = 12)
Age *mean* (SD)	47.1 (11.0)	52.9 (13.7)	54.8 (16.3)	42.3 (13.9)
Male *n* (%)	7 (64)	3 (33)	1 (17)	8 (69)
Years since surgery *mean* (range)	12.9 (5–21)	17.6 (8–28)	14.2 (3–30)	-
Daily bowel movements *mean* (range)	5.5 (3–8)	11.2 (5–20)	5.5 (1–8)	1.2 (1–2)
cPDAI score *mean* (range)	0.6 (0–1)	3.7 (3–5)	1.0 (1–1)	-
Continues antibiotics *n* (%)	0 (0)	3 (33)	0 (0)	0 (0)

cPDAI, clinical Pouchitis Disease Activity Index; FAP, familial adenomatous polyposis; HCs, healthy controls.

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
