# Peer review of "The Microbiota Profile in Inflamed and Non-Inflamed Ileal Pouch–Anal Anastomosis"

_microorganisms, 2020, doi:10.3390/microorganisms8101611_

Round 1

Reviewer 1 Report

Kousgaard and colleagues set out to study the microbiota composition of inflamed and non-inflamed pouches of patients with IPAA.

Overall Strengths

-Access to material from patients with a non-inflamed pouch, including those with a pouch due to FAP rather than IBD, allows the authors to better tease apart the factors associated with microbiota composition

-The use of PDAI to grade pouchitis is valuable

Overall Weaknesses

-The weaknesses of this study are very well outlined by the authors in their discussion, and I would like to commend them for including this section.

-The number of patients is low. Recruitment can be a challenge, but the low number of patients makes it difficult to draw firm conclusions

-The use of antibiotics by some patients also confounds interpretation of the data

-The lack of UC patients without a pouch means it is impossible to determine if the effects seen on the microbiota are due to the combination of factors, or specifically relating to pouchitis

Specific Comments

-More discussion of the fact that FAP and chronic pouchitis patient similarity in terms of microbial diversity and richness is warranted (Figures 1A and 1B). This is potentially the most striking/interesting finding of the paper. This suggests that it is not the inflammation of the pouch per se that is the issue as would be suggested by comparing patients with normal pouch and chronic pouchitis.

-In their discussion, the authors state that "it is important to take into consideration that patients with a normal functioning pouch
still have lower microbial diversity and richness, compared to healthy individuals with a colon". If the authors are referring to another study they should cite here. If referring to their own study, while it appears to be true when visually assessing the data in Figures 1A and 1B, no statistical support is provided for this statement.

-Figure 2B: What exactly is this data...UniFrac? Is it weighted or unweighted? Have the authors determined statistically that these groups are distinct (e.g. by comparing UniFrac distances). They appear to be, but statistical support should be provided.

Author Response

Response to Reviewer 1 Comments:

Kousgaard and colleagues set out to study the microbiota composition of inflamed and non-inflamed pouches of patients with IPAA.

Point 1:

Overall Strengths

-Access to material from patients with a non-inflamed pouch, including those with a pouch due to FAP rather than IBD, allows the authors to better tease apart the factors associated with microbiota composition

-The use of PDAI to grade pouchitis is valuable

Response 1: 

Thank you for the comments. 

Overall Weaknesses

-The weaknesses of this study are very well outlined by the authors in their discussion, and I would like to commend them for including this section.

Point 2:

-The number of patients is low. Recruitment can be a challenge, but the low number of patients makes it difficult to draw firm conclusions

Response2:

We acknowledge that the low number of patients is a study weakness making it difficult to draw conclusions. We have elaborated this in the discussion.

P8L272: added “… which makes it difficult to draw definitive conclusions, especially for FAP patients”.

Point 3:

-The use of antibiotics by some patients also confounds interpretation of the data

Response 3:

The use of antibiotics by some patients will inevitably affect the microbial results and confounds the interpretation of the data. This is already mentioned in the discussion.

Point 4:

-The lack of UC patients without a pouch means it is impossible to determine if the effects seen on the microbiota are due to the combination of factors, or specifically relating to pouchitis

Response 4:

It is a weakness that our study did not include UC patients with an intact colon, to compare inflamed pouch microbiota with inflamed colon microbiota and determine if the changes in the microbiota are due to the inflammation or a combination of factors. This is already mentioned in the discussion.

Specific Comments

Point 5:

-More discussion of the fact that FAP and chronic pouchitis patient similarity in terms of microbial diversity and richness is warranted (Figures 1A and 1B). This is potentially the most striking/interesting finding of the paper. This suggests that it is not the inflammation of the pouch per se that is the issue as would be suggested by comparing patients with normal pouch and chronic pouchitis.

Response 5:

We agree that this is an interesting finding and we have now elaborated on it in the discussion. However, it is difficult to interpret on this finding. Furthermore, one should note that the group of FAP patients only consist of six patients. We encourage future studies to investigate this further.

P7L226-227: added “This could suggest that factors other than changes in the microbial composition, can influence inflammation of the pouch”.

Point 6:

-In their discussion, the authors state that "it is important to take into consideration that patients with a normal functioning pouch still have lower microbial diversity and richness, compared to healthy individuals with a colon". If the authors are referring to another study they should cite here. If referring to their own study, while it appears to be true when visually assessing the data in Figures 1A and 1B, no statistical support is provided for this statement.

Response 6:

Figure 1A and 1B are now updated with p-values for the difference in microbial diversity and richness between healthy individuals and patients with a normally functioning pouch. The p-values are also provided in the text (P4L170).

Point 7:

-Figure 2B: What exactly is this data...UniFrac? Is it weighted or unweighted? Have the authors determined statistically that these groups are distinct (e.g. by comparing UniFrac distances). They appear to be, but statistical support should be provided.

Response 7:

We have now provided a more detailed description of the data analysis to Figure 2B.

P3-4L140-144: added “Beta diversity was examined using principal component analysis (PCA) on Hellinger transformed ASV abundances. Filtering of ASVs with low variance, defined as >50% of samples associated with one value, were performed prior to PCA. To asses statistical significance of the groupings in PCA, permutation tests of pairwise linear regression was performed using the pairwise.factorfit function from the RVAideMemoire package [27]”.

The distinct difference between the groups of participants is now provided in the manuscript.

P7L202: added “… all other groups (p < 0.05, Figure 2B)”.

P7L205-206: added “Albeit pairwise comparisons of patients with a normal pouch function, FAP and chronic pouchitis were not statistically significant (p-value between 0.14 – 0.26)”.

Reviewer 2 Report

This is an interesting study that compared microbial diversity and richness among patients with and without pouchitis. As the authors admitted, one important control group, UC patients with intact colon is missing, which compromises the quality of this study. Another possible control group would be UC patients after Hartmann procedure (before the creation of pouch).

The FAP patients apparently did not have pouchitis, however, the microbial profile is similar to UC patients with chronic pouchitis, and significantly different from patients without pouchitis. This needs to be explained.

Chronic pouchitis patients have frequent bowel movement. Could the altered microbial profile be explained by frequent bowel movement that affects the establishment of relevant bacterial colonies?

The authors mentioned antibiotics as a probable cofounding factor. The patients may also be on other medications. It may be interesting to explore the impact of other medications.

Author Response

Response to Reviewer 2 Comments:

Point 1:

This is an interesting study that compared microbial diversity and richness among patients with and without pouchitis. As the authors admitted, one important control group, UC patients with intact colon is missing, which compromises the quality of this study. Another possible control group would be UC patients after Hartmann procedure (before the creation of pouch).

Response 1:

Thank you for the comment. We acknowledge that a control group of UC patients and patients undergoing different surgical procedure e.g. Hartmann procedure could be relevant to include in the study, and this would be relevant for future studies.

Point 2:

The FAP patients apparently did not have pouchitis, however, the microbial profile is similar to UC patients with chronic pouchitis, and significantly different from patients without pouchitis. This needs to be explained.

Response 2:

We agree that this is an interesting finding and we have now elaborated on it in the discussion. However, it is difficult to interpret on this finding. Furthermore, one should note that the group of FAP patients only consist of six patients. We encourage future studies to investigate this further.

P7L226-227: added “This could suggest that factors other than changes in the microbial composition, can influence inflammation of the pouch”.

Point 3:

Chronic pouchitis patients have frequent bowel movement. Could the altered microbial profile be explained by frequent bowel movement that affects the establishment of relevant bacterial colonies?

Response 3:

An increase in bowel movements could perhaps influence the microbial profile. A study from Kwon et al found that microbial richness tended to decrease with increasing number of defecations in healthy individuals with different frequency of bowel movements. We have now included this in the discussion.

P7L227-229: added “Finally, frequent bowel movements may impact the microbial profile. A study by Kwon et al [30], found that microbial richness tended to decrease with increasing number of defecations in a group of healthy individuals”.

Point 4:

The authors mentioned antibiotics as a probable cofounding factor. The patients may also be on other medications. It may be interesting to explore the impact of other medications.

Response 4:

We agree that other medications could influence the gut microbiota and therefore could be a cofounding factor in our study. However, it would require a large number of patients to detect an effect of different medications on the microbiota.

This is now included in the discussion.

P8L278-280: added “Moreover, other types of medications could also influence the gut microbiota and would have been relevant to include in our study”.

Round 2

Reviewer 2 Report

na